# An In Vitro Assessment Method for Chemotherapy-Induced Peripheral Neurotoxicity Caused by Anti-Cancer Drugs Based on Electrical Measurement of Impedance Value and Spontaneous Activity

**DOI:** 10.3390/pharmaceutics15122788

**Published:** 2023-12-16

**Authors:** Xiaobo Han, Naoki Matsuda, Yuto Ishibashi, Mikako Shibata, Ikuro Suzuki

**Affiliations:** Department of Electronics, Graduate School of Engineering, Tohoku Institute of Technology, 35-1 Yagiyama Kasumicho, Taihaku-ku, Sendai 9828577, Japan; xiaobohan@tohtech.ac.jp (X.H.); na-matsuda@tohtech.ac.jp (N.M.); yishibashi@tohtech.ac.jp (Y.I.); m-shibata@tohtech.ac.jp (M.S.)

**Keywords:** chemotherapy-induced peripheral neurotoxicity, anti-cancer drugs, microelectrode array, impedance value, spontaneous activity

## Abstract

Chemotherapy-induced peripheral neurotoxicity (CIPN) is a major adverse event of anti-cancer drugs, which still lack standardized measurement and treatment methods. In the present study, we attempted to evaluate neuronal dysfunctions in cultured rodent primary peripheral neurons using a microelectrode array system. After exposure to typical anti-cancer drugs (i.e., paclitaxel, vincristine, oxaliplatin, and bortezomib), we successfully detected neurotoxicity in dorsal root ganglia neurons by measuring electrical activities, including impedance value and spontaneous activity. The impedance value decreased significantly for all compounds, even at low concentrations, which indicated cell loss and/or neurite degeneration. The spontaneous activity was also suppressed after exposure, which suggested neurotoxicity again. However, an acute response was observed for paclitaxel and bortezomib before toxicity, which showed different mechanisms based on compounds. Therefore, MEA measurement of impedance value could provide a simple assessment method for CIPN, combined with neuronal morphological changes.

## 1. Introduction

Chemotherapy-induced peripheral neuropathy (CIPN) is a major common adverse event and debilitating toxicity associated with cancer therapy [1]. The number of cancer survivors is projected to increase substantially over the next few decades due to advances in the early diagnosis and treatment of cancers. Consequently, the proportion of cancer patients suffering from CIPN is also expected to increase [2]. CIPN is primarily associated with neurological abnormalities linked to pain, loss of sensation, and motor functionality, ultimately leading to a decreased quality of life [1,2,3,4,5]. Classes of cancer therapy implicated in CIPN include platinums, taxanes, vinca alkaloids, proteosome inhibitors, and angiogenesis inhibitors [5,6,7]. Recent research has also included many efforts to synthesize derivatives of these drugs with more efficient, selective, and less toxic properties [8,9,10]. Despite the high prevalence and morbidity associated with CIPN, the diagnosis and treatment of CIPN remain challenging because its clinical presentation and molecular mechanisms are heterogeneous, and there is no standardized measure for CIPN yet [11,12]. An accurate assessment is essential to improve knowledge about CIPN incidence.

Recently, in vitro cell models of primary rodent dorsal root ganglia (DRG) sensory neurons have been developed to study CIPN based on cell viability and morphology analysis [13,14,15,16]. For example, both bortezomib- and vincristine-treated neurons reportedly showed decreased neurite outgrowth without increased cell death [17], while cisplatin and oxaliplatin treatment induced cell death [18]. Therefore, cultured DRG neurons can serve as a reliable and robust in vitro model for mechanistic and therapeutic CIPN studies.

In the present study, we applied cultured DRG neurons for microelectrode array (MEA) measurement to gather meaningful data pertaining to peripheral neuropathy. MEA provides noninvasive and real-time measurements, which have been utilized with varied success in both in vivo and in vitro neural electrophysiological evaluation, including for DRG neurons [19,20,21,22,23]. Traditionally, electrical information is analyzed after MEA measurement in the form of voltage. The impedance value is another critical feature during electrical measurement, which, however, has been relatively less explored in previous MEA-related studies. Measurement of the impedance could show information about cells interfacing with the MEA surface, probably providing a more direct relationship with cell morphology. Here, we demonstrate that consistent impedance measurements can be obtained from DRG neurons seeded on a MEA plate. Then, changes in the value of impedance could reflect DRG response to representative anti-cancer drugs (i.e., paclitaxel, vincristine, oxaliplatin, and bortezomib) and be used for predicting neurotoxicity. Combined with the measurement of spontaneous activity, this MEA measurement with cultured DRG neurons has the potential for in vitro CIPN prediction.

## 2. Materials and Methods

### 2.1. Culture of Primary DRG Neurons

Primary rodent DRG neurons were harvested and cultured as described previously [18]. Briefly, DRG neurons were collected from embryos of one timed-pregnant (E14) Sprague Dawley rats (total ~10 embryos). All procedures were performed according to the Guide for the Care and Use of Laboratory Animals published by the US National Institutes of Health [24] and were approved by the Tohoku Institute of Technology Animal Care and User Committee. Firstly, the rats were asphyxiated with isoflurane, and embryos were recollected. Then, spinal cords with DRGs were carefully isolated and removed from embryos. After plucking off DRGs from spinal cords, the sensory neurons were dissociated by incubation for 30 min with 0.25% Trypsin at 37 °C. After cell counting, approximately 5.0 × 10^4^ dissociated DRG neurons (6.0 × 10^5^ cells/cm^2^) in 10 µL Neurobasal neuronal medium (with B-27 supply, Gibco, Grand Island, NY, USA) were seeded directly into the MEA plate. After 1 h, another 600 µL of Neurobasal neuronal medium was applied. The next day, the medium was replaced with 600 µL of Neurobasal neuronal medium containing 1 µM ara-C kept for 3 days to suppress the proliferation of glial cells. Afterward, the medium was changed back to 600 µL Neurobasal neuronal medium, and half the volume of the medium was replaced twice per week.

### 2.2. Extracellular Recording

Measurement of impedance and spontaneous extracellular field potentials were acquired at 37 °C under a 5% CO_2_ atmosphere using an MEA system (Maestro Pro, Axion Biosystems, Atlanta, GA, USA). After setting MEA plates, impedance of all electrodes was tracked and obtained for 3 min. Then, spontaneous activities were recorded for 5 min at a sampling rate of 10 kHz/channel. All signals were stored on a personal computer.

After cultured DRG neurons for 14 days, four representative anticancer drugs were administered to the cultures at two different concentrations (low and high) each: paclitaxel at 0.1 µM (n = 4 wells) and 1 µM (n = 5 wells), vincristine at 0.003 µM (n = 4 wells) and 0.03 µM (n = 5 wells), oxaliplatin at 10 µM (n = 4 wells) and 100 µM (n = 5 wells), and bortezomib at 0.001 µM (n = 4 wells) and 0.01 µM (n = 5 wells). Sucrose (10 µM) was added as a negative drug (n = 5 wells), and DMSO (0.1%) as a control drug to the cultures (n = 5 wells). The drug exposure lasted for 168 h at 37 °C. Before drug exposure (before) and 24 h, 72 h, and 168 h after exposure, measurements of impedance and spontaneous activities were performed.

### 2.3. Immunocytochemistry

The sample cultures after MEA measurements at 168 h were fixed with 4% paraformaldehyde in PBS on ice (4 °C) for 10 min. Fixed cells were incubated with 0.2% Triton-X-100 in PBS for 5 min, then with preblock buffer (0.05% Triton-X and 5% FBS in PBS) at 4 °C for 1 h, and finally with preblock buffer containing a specific primary antibody, mouse anti-β-tubulin III (1:1000, T8578, Sigma–Aldrich, St. Louis, MO, USA), at 4 °C for 24 h. The samples were then incubated with a secondary antibody, anti-mouse 488 Alexa Fluor (1:1000 in preblock buffer, ab150113, Abcam, Waltham, MA, USA), for 1 h at room temperature. Stained cultures were washed twice with preblock buffer and rinsed twice with PBS. A confocal microscope (Eclipse Ti2-U, Nikon, Tokyo, Japan) was used to capture local images, and then ImageJ software (Version 1.54g, NIH) was used to adjust image intensity.

### 2.4. Statistics

A one-way analysis of variance (ANOVA) followed by a Dunnett test was used to calculate significant differences between the compounds at different time points in impedance values and total spikes. For two-dimensional scatter plots of impedance and Coefficient of Variation of impedance, a one-way multivariate analysis of variance (MANOVA) was used to calculate significant differences between the compounds and negative controls.

## 3. Results

### 3.1. Morphological and Impedance Changes of Cultured DRG Responding to Anti-Cancer Drugs

Figure 1 shows representative immunofluorescence images and time-order impedance changes of cultured DRG neurons after 168 h of exposure to negative controls or anti-cancer drugs. Under negative or control drugs, neurites grew sufficiently to occupy almost the whole MEA area, and impedance values were maintained at a similar level at about 40 kΩ for all electrodes throughout 168 h of exposure.

After exposure tofour typical anti-cancer drugs at low concentrations, no obvious morphological changes can be observed from immunofluorescence images. However, impedance values of most electrodes decreased after 168 h of exposure, except for bortezomib. After exposure to paclitaxel at 1 µM and bortezomib at 0.01 µM, a fragmented appearance of axonal fibers was observed in a wide range of areas from immunofluorescence images. After exposure to vincristine at 0.03 µM, significantly enlarged holes were observed among neurites. There were still no obvious morphological changes after exposure to oxaliplatin at 100 µM. After 72 h exposure to vincristine, oxaliplatin, and bortezomib at high concentrations, an obvious decrease in impedance was observed for most electrodes. This decrease became more severe after 168 h exposure. These results indicated that the impedance value could reflect DRG neuron responses to anti-cancer drugs, which correspond to morphological changes. DRG responses to compound administration could be detected at low concentrations based on impedance value changes.

### 3.2. Impedance and Spontaneous Measurements Reflect DRG Neurons’ Response to Anti-Cancer Drugs

To develop an effective assessment method for peripheral neurotoxicity by electrical parameters, we further measured the spontaneity of DRG under exposure to anti-cancer drugs and evaluated the results together with impedance values. After MEA measurements, the impedance value, total spikes, Coefficient of Variation (CV) of the impedance value, and CV of total spikes were calculated. Figure 2 shows the calculated results of these four parameters and distribution maps of impedance/total spike changes on each electrode of measured wells at different time points after exposure to control and negative drugs. The calculated results of electrical parameters and corresponding distribution maps after exposure to paclitaxel, vincristine, oxaliplatin, and bortezomib are sequentially exhibited in Figure 3, Figure 4, Figure 5 and Figure 6.

After exposure to control or negative drugs, the impedance value showed almost no change at each time point. Total spikes showed vibration at different time points but without significant differences. The CV of impedance increased time-dependently after exposure to sucrose, which means that few electrodes had a quite high impedance, as shown in the distribution map. The impedance value significantly decreased after 168 h of exposure to paclitaxel at both 0.1 μM and 1 µM. Meanwhile, the CV of impedance significantly increased, which indicated that this impedance decreased, focused on several electrodes—but not all—as shown in the distribution maps. After exposure to 0.1 µM paclitaxel, total spikes showed a tendency to decrease along with the exposure time. However, total spikes increased after 24 h of exposure to 1 µM of paclitaxel and then decreased till elimination. After exposure to 0.003 µM of vincristine, the impedance value significantly decreased after 72 h and 168 h, with a significant increase in CV of impedance at 168 h. Though the same tendency was observed in samples exposed to 0.03 µM vincristine, there was no significant difference. An increasing total spike was also observed after 24 h of exposure to 0.03 µM of vincristine. The impedance value significantly decreased after 168 h of exposure to 10 µM of oxaliplatin, and it became severe for samples exposed to 100 µM of oxaliplatin, with an earlier significant decrease at 72 h. Total spikes decreased till elimination along with exposure time for both 10 µM and 100 µM of oxaliplatin. The impedance value showed almost no change at different time points after exposure to 0.001 µM of bortezomib. However, the impedance value significantly decreased after 168 h of exposure to 0.01 µM of bortezomib, which indicated a dose-dependent reaction manner. For both samples exposed to 0.001 or 0.01 µM of bortezomib, total spikes increased at 24 h, followed by a dramatic decrease to elimination.

## 4. Discussion

The impedance value is a critical feature of electrode arrays, and nevertheless, the impedance will increase when cells contact electrodes. This provides the possibility of constructing a simple platform using impedance as a tool to evaluate neuron–electrode interface conditions. Previous research also reported such an approach to monitor neurite outgrowth from cultured DRG neurons [25]. In the present study, we demonstrate a method to predict peripheral toxicity via impedance measurement, which was never reported before. We verified the current assessment method with four typical anti-cancer drugs (i.e., paclitaxel, vincristine, oxaliplatin, and bortezomib) and showed convincing results. Combined with the measurement of spontaneous activity, it is possible to predict that peripheral neuropathy occurred through different mechanisms.

MEA measurements of electrical activities were conducted with cultured DRG neurons after the administration of representative anti-cancer drugs that are widely reported to induce peripheral neuropathy in clinical use [6,7]. The mechanistic basis for CIPN development induced by these drugs is still less clear, even though several groups have reported drug-induced neurotoxicity based on in vitro analysis of cell viability and/or morphology. For example, paclitaxel toxicity was confirmed with neurite degeneration in cultured primary DRG neurons after 1 µM of exposure for 24 h [26]. Such morphological changes in neurites were not significant after 168 h of exposure in the present study, which may be attributed to the high cell-seeding density on MEA plates compared to previous research. However, the influence of paclitaxel could be detected on DRG neurons by a significant decrease in impedance values, even at a lower concentration (0.1 μM) compared to previous research. Additionally, we also performed measurements of spontaneous activity after paclitaxel administration, and interestingly, we observed an acute increase in total spikes, followed by a decrease and even elimination. It is reported that paclitaxel could increase the porosity of membranes, leading to hyperexcitability that has been observed at pre-toxic conduction [27,28], which could explain our finding. Many of the effects of vincristine are described as similar to those of paclitaxel [6]. Vincristine exposure greater than 0.03 μM could induce axonal degeneration in cultured primary DRG neurons [15,17], which could explain the observed impedance decreasing in the present study. Furthermore, the impedance value showed a time-dependent decrease after exposure to vincristine at 0.003 μM, suggesting a potential for toxicity at a low concentration. The accumulation of oxaliplatin and its metabolite in DRG and the formation of platinum-DNA adducts are considered key steps in neurotoxicity development [29,30,31]. Previous in vitro studies also show cell apoptosis and neuron loss in DRG after oxaliplatin administration at high doses [18,30,31], which could explain the observed impedance decreasing in the present study. Oxaliplatin could also alter voltage-gated ion channel expression in DRG [16,32], which induced a decrease in spontaneous spikes. Bortezomib is known to induce various forms of neuronal toxicity in DRG, including mitochondria injury, endoplasmic reticulum stress, and neurite outgrowth decreases [17,33,34,35]. However, it is reported that bortezomib administration at a low concentration did not affect DRG morphology [36], which aligns with our results showing no significant change in impedance values. Moreover, bortezomib could induce increases in metabolic activity at a low concentration before a subsequent decrease [36,37], which agreed with our results showing a significant increase in spontaneous spikes after 24 h of poration exposure.

Taken together, the current study has demonstrated a consistent and time-dependent decrease in the impedance value in cultured DRG neurons following the administration of anti-cancer drugs (Figure 7). We calculated significant differentiation between each compound with negative controls (Table 1). The outcomes revealed significant differences among nearly all anti-cancer drugs after 168 h of exposure. Significant differences were not detected in 0.001 μM of bortezomib but were in 0.01 μM of bortezomib, indicating a dose-dependent response, the same as previous reports [17,35]. Significant differences were observed in 0.003 μM of vincristine but not in 0.03 μM of vincristine. Similarly, previous studies investigating the impact of dose and peak plasma concentration on vincristine-induced CIPN have yielded inconclusive results [38,39]. These findings suggested that the impedance value can be considered a reliable parameter for predicting peripheral neurotoxicity induced by these drugs.

Interestingly, the administration of paclitaxel and vincristine elicited a similar trend, characterized by a time-dependent decrease in impedance and a concomitant increase in the CV of impedance. This specific change pattern was not observed in DRG after the administration of oxaliplatin or bortezomib. Since vincristine has similar effects as paclitaxel on DRG [6], this finding suggested that a combination of the impedance value with the CV of impedance could serve as a valuable tool for detecting certain types of peripheral toxicity with similar underlying mechanisms. However, it is essential to emphasize that this is a preliminary finding, and further research and validation using more anti-cancer drugs with varying mechanisms are necessary to establish this method as a valuable and reliable tool. To understand the observed impedance changes, it is crucial to evaluate the relationship between the impedance value and neuronal morphology (refer to Figure 1). However, one complicating factor was the substantially higher cell culture density in MEA plates compared to previous in vitro cultures, making it challenging to conduct further quantitative morphological analyses using traditional methods, such as manual labeling [40]. To address this challenge, our group recently developed a morphological deep learning analysis method specifically designed to analyze soma and axonal images separately [41]. This method has enabled us to identify and quantify morphological changes induced by anti-cancer drugs at low concentrations, followed by the prediction of mechanisms. The results obtained from this morphological deep learning analysis provide valuable insights into the mechanisms underlying the observed impedance decreases in the current study. With the current MEA measurement system, it is possible to explore the correlation between impedance values and neuronal degeneration within the same platform while leveraging advanced image analysis techniques to gain a more comprehensive understanding of the effects of anti-cancer drugs on DRG neurons.

## 5. Conclusions

In this study, we developed an in vitro assessment method for predicting CIPN using impedance value and spontaneous activity. The peripheral neurotoxicity of anti-cancer drugs was successfully detected by a time-dependent decrease in impedance value, even at low concentrations. Furthermore, specific change patterns in impedance value and spontaneous activity were observed in certain compounds. These observations contribute to the understanding of the mechanisms underlying CIPN development and emphasize the potential of electrical parameters, such as impedance and spontaneous activity, as valuable indicators for the early detection and assessment of peripheral neurotoxicity. In summary, this study contributes to advancing our understanding of CIPN mechanisms and highlights the utility of in vitro electrical assessments for predicting and assessing peripheral neurotoxicity. This research could have implications for the development of safer and more effective anti-cancer therapies by allowing for the early identification of potential neurotoxic effects during the drug-development process.

## Figures and Tables

**Figure 1 pharmaceutics-15-02788-f001:**
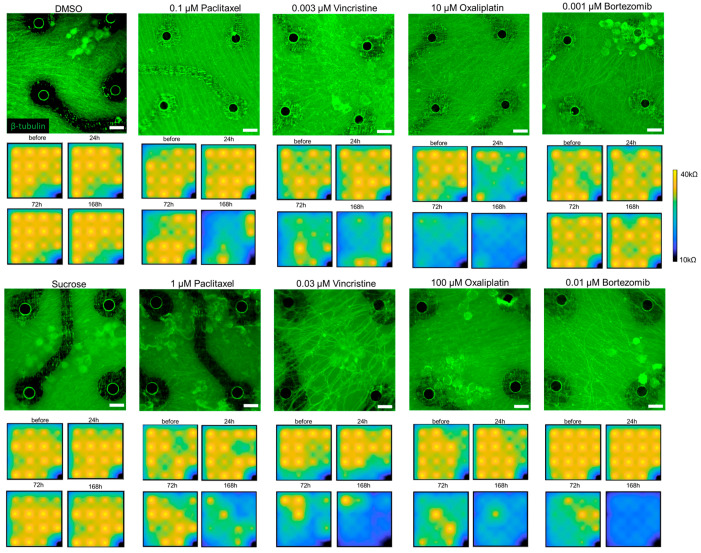
Representative immunofluorescence images of cultured DRG neurons after 168 h of exposure and corresponding impedance color maps before or after 24 h, 72 h, and 168 h exposure to DMSO (vehicle), sucrose (negative), 0.1 µM and 1 µM of paclitaxel, 0.003 µM and 0.03 µM of vincristine, 10 µM and 100 µM of oxaliplatine, and 0.001 µM and 0.01 µM of bortezomib. In immunofluorescence images, green is β-tubulin, and scale bar = 20 µm.

**Figure 2 pharmaceutics-15-02788-f002:**
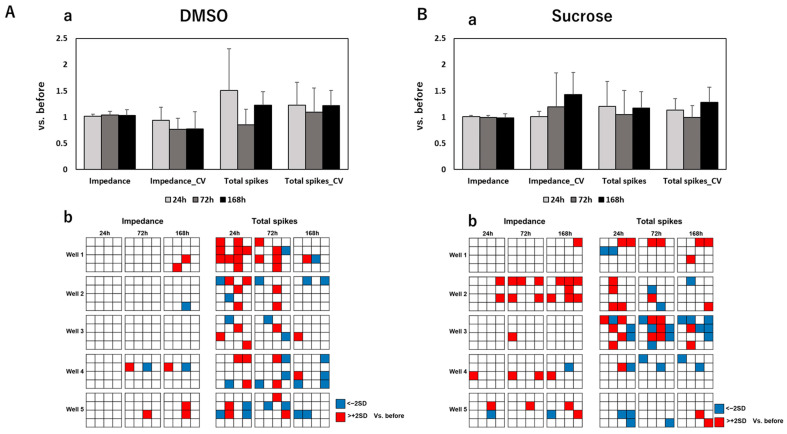
Total and individual changes of four electrical parameters (i.e., the impedance value, Coefficient of Variation (CV) of impedance value, total spikes, and CV of total spikes) in DRG after 24 h, 72 h, and 168 h of exposure to (**A**) control drug, DMSO, and (**B**) negative drug, sucrose. The upper column (**a**) shows totally fold changes of four parameters compared to that before drug administration; data are shown as mean + standard deviation (SD), n = 5 MEA plates for both DMSO and sucrose. The lower column (**b**) shows the distribution maps of the impedance value changes and total spike changes in each individual MEA plate at different time points. One table represents an MEA plate with 16 electrodes, and one grid in the table represents 1 electrode in the MEA plate; electrodes with an increase in value greater than 2 times the SD compared to the value before drug administration are marked as red, while electrodes with a decrease in value greater than 2 times the SD are marked as blue.

**Figure 3 pharmaceutics-15-02788-f003:**
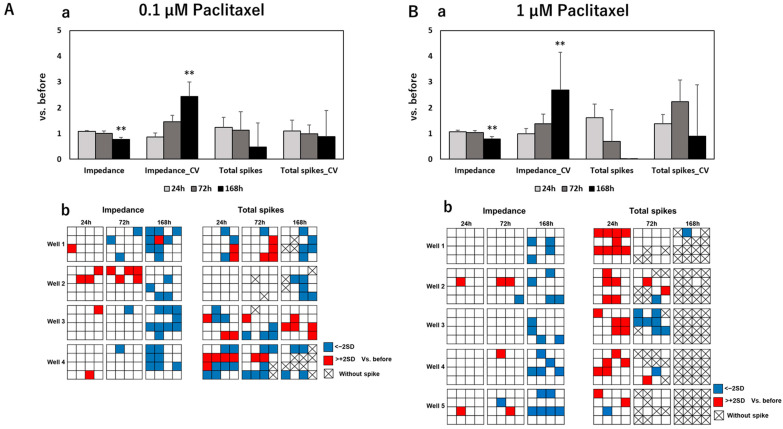
Total and individual changes of four electrical parameters (i.e., the impedance value, CV of impedance value, total spikes, and CV of total spikes) in DRG after 24 h, 72 h, and 168 h of exposure to (**A**) 0.1 µM of paclitaxel and (**B**) 1 µM of paclitaxel. The upper column (**a**) shows total fold changes of four parameters compared to that before drug administration; data are shown as mean + SD, n = 4 for 0.1 µM of paclitaxel and n = 5 for 1 µM of paclitaxel. ** *p* < 0.01 vs. before. The lower column (**b**) shows the distribution maps of the impedance value changes and total spike changes in each individual MEA plate at different time points with same labels as in Figure 2. The spontaneous spike could not be detected in some electrodes after drug administration, which is shown as × in the grid.

**Figure 4 pharmaceutics-15-02788-f004:**
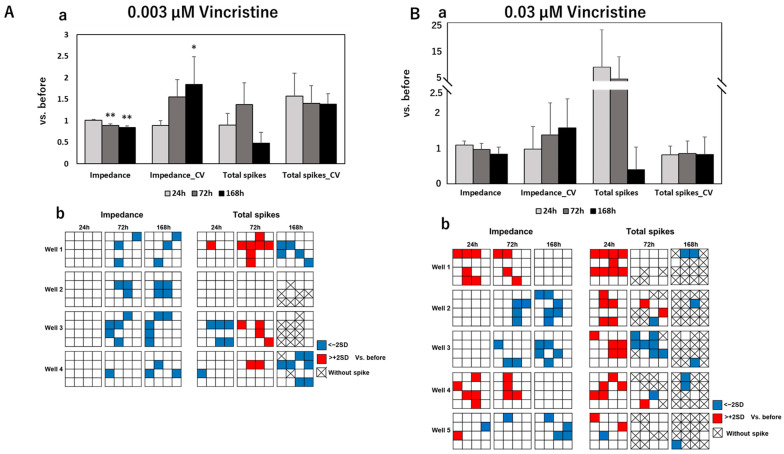
Total and individual changes of the impedance value, CVof impedance value, total spikes, and CV of total spikes in DRG after 24 h, 72 h, and 168 h of exposure to (**A**) 0.003 µM of vincristine and (**B**) 0.03 µM of vincristine. The upper column (**a**) shows fold changes of four parameters compared to that before drug administration; data are shown as mean + SD, n = 4 for 0.003 µM of vincristine and n = 5 for 0.03 µM of vincristine. * *p* < 0.05 vs. before; ** *p* < 0.01 vs. before. The lower column (**b**) shows the distribution maps of the impedance value changes and total spike changes in each individual MEA plate at different time points with same labels as in Figure 3.

**Figure 5 pharmaceutics-15-02788-f005:**
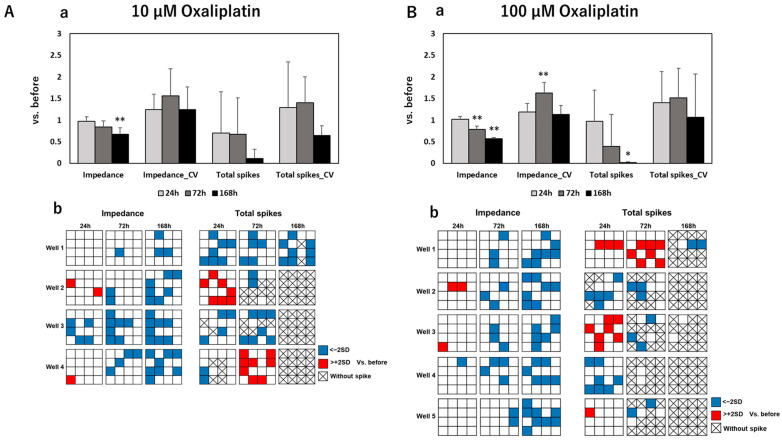
Total and individual changes of the impedance value, CV of impedance value, total spikes, and CV of total spikes in DRG after 24 h, 72 h, and 168 h of exposure to (**A**) 10 µM of oxaliplatin and (**B**) 100 µM of oxaliplatin. The upper column (**a**) shows fold changes of four parameters compared to that before drug administration; data are shown as mean + SD, n = 4 for 10 µM of oxaliplatin and n = 5 for 100 µM of oxaliplatin. * *p* < 0.05 vs. before; ** *p* < 0.01 vs. before. The lower column (**b**) shows the distribution maps of the impedance value changes and total spike changes in each individual MEA plate at different time points with same labels as in Figure 3.

**Figure 6 pharmaceutics-15-02788-f006:**
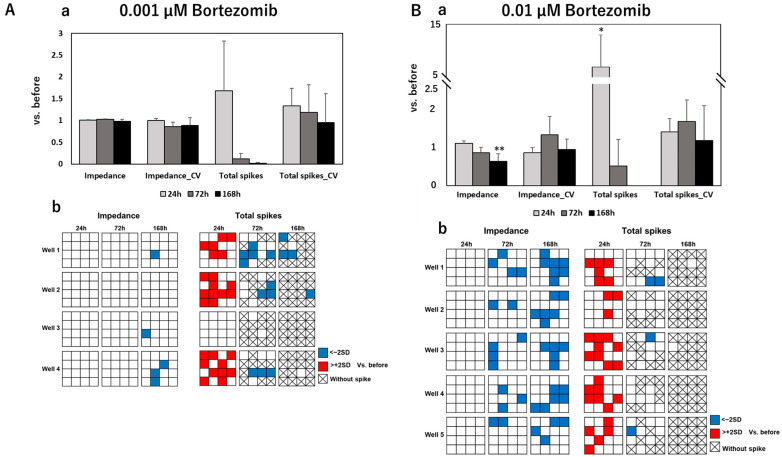
Total and individual changes of the impedance value, CV of impedance value, total spikes, and CV of total spikes in DRG after 24 h, 72 h, and 168 h of exposure to (**A**) 0.001 µM of bortezomib and (**B**) 0.01 µM of bortezomib. The upper column (**a**) shows fold changes of four parameters compared to that before drug administration; data are shown as mean + SD, n = 4 for 0.001 µM of bortezomib and n = 5 for 0.01 µM of bortezomib. * *p* < 0.05 vs. before; ** *p* < 0.01 vs. before. The lower column (**b**) shows the distribution maps of the impedance value changes and total spike changes in each individual MEA plate at different time points with same labels as in Figure 3.

**Figure 7 pharmaceutics-15-02788-f007:**
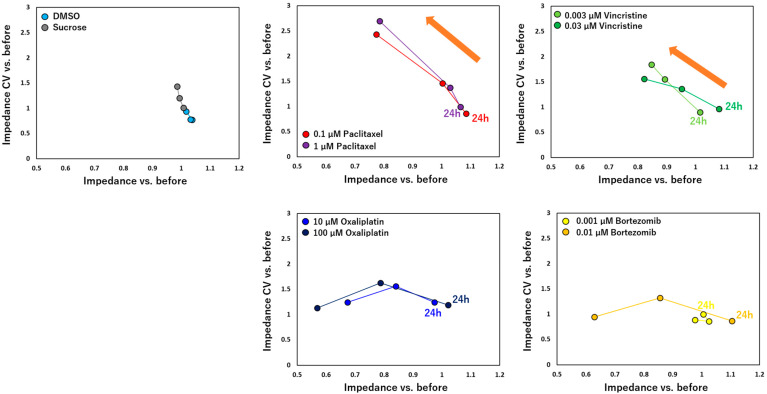
Scatter plots of fold changes in impedance (*x* axis) and CV of impedance (*y* axis) at each time point after drug administrations compared to before. After administration of paclitaxel and vincristine at both low and high concentrations, time-dependent decrease in impedance and concomitant increase in CV of impedance were confirmed. Plots after administration of paclitaxel and vincristine show the same trend, indicated by orange arrows.

**Table 1 pharmaceutics-15-02788-t001:** One-way multivariate analysis of variance results between each compound compared to negative controls.

Compounds Administration Conditions		vs. DMSO			vs. Sucrose		
		24 h	72 h	168 h	24 h	72 h	168 h
DMSO	24 h	–	*p* = 0.54	*p* = 0.61	*p* = 0.86	*p* = 0.68	*p* = 0.14
	72 h	*p* = 0.54	–	*p* = 0.99	*p* = 0.15	*p* = 0.40	* *p* = 0.043
	168 h	*p* = 0.61	*p* = 0.99	–	*p* = 0.28	*p* = 0.51	*p* = 0.09
Sucrose	24 h	*p* = 0.86	*p* = 0.15	*p* = 0.28	–	*p* = 0.77	*p* = 0.11
	72 h	*p* = 0.68	*p* = 0.40	*p* = 0.51	*p* = 0.77	–	*p* = 0.81
	168 h	*p* = 0.14	* *p* = 0.043	*p* = 0.09	*p* = 0.11	*p* = 0.81	–
0.1 µM of paclitaxel	24 h	*p* = 0.086	*p* = 0.29	*p* = 0.33	** *p* = 0.0063	* *p* = 0.027	* *p* = 0.016
	72 h	** *p* = 0.0071	** *p* = 0.0025	** *p* = 0.0057	** *p* = 0.0039	*p* = 0.64	*p* = 0.95
	168 h	** *p* = 0.0024	** *p* = 0.0024	** *p* = 0.0058	** *p* = 0.0012	** *p* = 0.0037	** *p* = 0.0097
1 µM of paclitaxel	24 h	*p* = 0.23	*p* = 0.12	*p* = 0.15	*p* = 0.27	*p* = 0.22	*p* = 0.075
	72 h	*p* = 0.13	* *p* = 0.049	*p* = 0.063	*p* = 0.16	*p* = 0.54	*p* = 0.71
	168 h	** *p* = 0.0010	** *p* = 0.0014	** *p* = 0.0054	** *p* = 0.58 × 10^−4^	** *p* = 0.0025	** *p* = 0.0059
0.003 µM of vincristine	24 h	*p* = 0.84	*p* = 0.63	*p* = 0.78	*p* = 0.34	*p* = 0.59	*p* = 0.073
	72 h	** *p* = 0.0076	* *p* = 0.025	*p* = 0.066	** *p* = 0.0031	* *p* = 0.012	*p* = 0.21
	168 h	** *p* = 0.0030	** *p* = 0.0092	* *p* = 0.036	** *p* = 0.69 × 10^−4^	** *p* = 0.0032	** *p* = 0.049
0.03 µM of vincristine	24 h	** *p* = 0.0070	*p* = 0.075	*p* = 0.12	* *p* = 0.018	*p* = 0.18	*p* = 0.31
	72 h	*p* = 0.61	*p* = 0.39	*p* = 0.39	*p* = 0.70	*p* = 0.88	*p* = 0.81
	168 h	*p* = 0.16	*p* = 0.10	*p* = 0.14	*p* = 0.18	*p* = 0.27	*p* = 0.34
10 µM of oxaliplatin	24 h	*p* = 0.41	*p* = 0.12	*p* = 0.17	*p* = 0.46	*p* = 0.92	*p* = 0.77
	72 h	*p* = 0.11	*p* = 0.079	*p* = 0.12	*p* = 0.11	*p* = 0.15	*p* = 0.26
	168 h	** *p* = 0.0077	* *p* = 0.010	* *p* = 0.019	** *p* = 0.0079	** *p* = 0.0091	* *p* = 0.021
100 µM of oxaliplatin	24 h	*p* = 0.11	* *p* = 0.023	* *p* = 0.043	*p* = 0.14	*p* = 0.70	*p* = 0.41
	72 h	** *p* = 0.0013	** *p* = 0.81 × 10^−5^	** *p* = 0.0056	** *p* = 0.24 × 10^−5^	** *p* = 0.0047	* *p* = 0.010
	168 h	** *p* = 2.71 × 10^−7^	** *p* = 1.01 × 10^−5^	** *p* = 6.16 × 10^−5^	** *p* = 1.01 × 10^−7^	** *p* = 3.44 × 10^−7^	** *p* = 5.45 × 10^−5^
0.001 µM of bortezomib	24 h	*p* = 0.90	*p* = 0.21	*p* = 0.39	*p* = 0.95	*p* = 0.82	*p* = 0.14
	72 h	*p* = 0.84	*p* = 0.79	*p* = 0.88	*p* = 0.11	*p* = 0.40	* *p* = 0.048
	168 h	*p* = 0.32	*p* = 0.45	*p* = 0.69	*p* = 0.30	*p* = 0.48	*p* = 0.14
0.01 µM of bortezomib	24 h	*p* = 0.10	*p* = 0.17	*p* = 0.19	* *p* = 0.019	* *p* = 0.039	** *p* = 0.0082
	72 h	*p* = 0.12	*p* = 0.11	*p* = 0.17	*p* = 0.13	*p* = 0.15	*p* = 0.20
	168 h	* *p* = 0.013	** *p* = 0.0090	* *p* = 0.019	* *p* = 0.012	* *p* = 0.019	* *p* = 0.025

*: *p* < 0.05, **: *p* < 0.01.

## Data Availability

The data and scripts that support the findings of this study are available from the corresponding author upon reasonable request.

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
