# Peer review of "An In Vitro Assessment Method for Chemotherapy-Induced Peripheral Neurotoxicity Caused by Anti-Cancer Drugs Based on Electrical Measurement of Impedance Value and Spontaneous Activity"

_pharmaceutics, 2023, doi:10.3390/pharmaceutics15122788_

Round 1

Reviewer 1 Report

Comments and Suggestions for Authors

Dear authors,

The manuscript entitled 'An in vitro assessment method for chemotherapy-induced peripheral neurotoxicity caused by anti-cancer drugs based on electrical measurement of impedance value and spontaneous activity' focuses on a crucial aspect of the adverse effect of anti-cancer treatment with specific drugs. 

 In my opinion only two aspect needs to be improved:

1. The goal of your article in the introduction should be extended. What are the possibilities of positive results?

2. From my perspective, it is a more important one. Why did you use DMSO as a vehicle? In clinic, the DMSO is never used as a solvent for drugs. 

Comments on the Quality of English Language

The manuscript is written with good quality English language.

Author Response

In my opinion only two aspect needs to be improved:

  1. The goal of your article in the introduction should be extended. What are the possibilities of positive results?

Thanks very much for your kind suggestion. To show the possibility of our results, a sentence was added in Page 6, line 60 as “This approach may contribute to a better understanding of neurotoxic effects associated with anti-cancer drugs and aid in the early prediction and evaluation of peripheral neuropathy.”

  1. From my perspective, it is a more important one. Why did you use DMSO as a vehicle? In clinic, the DMSO is never used as a solvent for drugs.

Thanks very much for your question. DMSO is commonly used as a vehicle in in vitro researches, since its high solubility and relatively low toxicity. In the present study, all drugs were dissolved well in DMSO, while DMSO itself did not influence electrical activity of DRG neurons. Therefore, we think that DMSO is a good vehicle for experiments using cultured neurons.

Reviewer 2 Report

Comments and Suggestions for Authors

1. Significant differences were not detected in 0.001 µM bortezomib, but in 0.01 µM bortezomib, indicating a dose-dependent

response same as previous reports.

Could you explain why the reason about this?

2. Significant differences were observed in 0.003 m Mama vincristine, but not in 0.03 µM vincristine.

Clarify this also.

3. Can you compare above two statment results.

4. Results maxima 168hrs. Is it any boundary about time. Why 168hrs maxi. In this investigation?

5. Abstract and conclusion mention the aim of this investigation.

The research and data seems intresting in the pharmaceutical field. Could be accepted after minor amendments.

Comments on the Quality of English Language

Its good.

Author Response

  1. Significant differences were not detected in 0.001 µM bortezomib, but in 0.01 µM bortezomib, indicating a dose-dependent response same as previous reports. Could you explain why the reason about this?

Thanks very much for your question. It is reported that a low concentration (0.005 µM) of bortezomib did not induce morphological changes in cultured DRG neuron [36]. While a high concentration (0.1 µM) of bortezomib induced neurites reduction [17]. Therefore, it is reasonable to consider that bortezomib influence cell morphology in a dose-dependent manner, which is same to our founding in impedance changes.

  1. Significant differences were observed in 0.003 µM vincristine, but not in 0.03 µM vincristine. Clarify this also.

Thanks very much for your question. Vincristine induce axonal injury from a low to high dose. And we also observed neurites reduction in DRG after 168h exposure to 0.03 µM vincristine (Figure 1). However, no significant difference in impedance changes was observed in 0.03 µM vincristine. Exactly, there is no existed data to support that there is a correlation between neuropathy and plasma concentrations of vincristine [39]. Besides concentrations, other factors might be necessary to explore for the influence on neuronal electrical activities. We would like to perform related experiments in future works.

  1. Can you compare above two statement results.

Thanks very much for your question. It seems interesting to compare the two above results of vincristine and bortezomib. It is reported that vincristine and bortezomib could induce same axonal injury, but through different upstream mechanisms [17]. While we observed neurites reduction in both vincristine and bortezomib at high concentrations, but the impedance change showed different tendency in the present study. This could be due to the different underlying mechanisms mentioned above, but more experimental results are necessary. We would like to perform more experiments about vincristine and bortezomib in future works, to explore the difference in these two drugs and compare our results to previous reports.

  1. Results maxima 168hrs. Is it any boundary about time. Why 168hrs maxi. In this investigation?

Thanks very much for your question. In the present study, significant changes in impedance were observed for most of drugs after 168h exposure, so we conclude results at 168h. Since cultured cells were still in good conditions from outlook as shown in Figure 1, 168h shall not be a boundary and experiments for longer time are possible. We would like to test more time points in the future to show the difference between each drug.

  1. Abstract and conclusion mention the aim of this investigation.

Thanks very much for your kind suggestion. To emphasize the aim of this investigation, a sentence was added in Page 11, Line 325, as “In summary, the study contributes to advancing our understanding of CIPN mechanisms and highlights the utility of in vitro electrical assessments for predicting and assessing peripheral neurotoxicity. This research could have implications for the development of safer and more effective anti-cancer therapies by allowing for the early identification of potential neurotoxic effects during the drug development process.”

Reviewer 3 Report

Comments and Suggestions for Authors

This work developed an in vitro assessment method for predicting CIPN using impedance values and spontaneous activity. Peripheral neurotoxicity of anticancer drugs was successfully detected by a decrease in impedance values over time, even at low concentrations. In addition, specific patterns of changes in impedance values and spontaneous activity were observed in certain compounds. These observations contribute to the understanding of the mechanisms underlying the development of CIPN and emphasise the potential of electrical parameters, such as impedance and spontaneous activity, as valuable indicators for early detection and assessment of peripheral neurotoxicity. I believe that this work requires minor modifications. Specific comments are as follows.

1. To make it easier for the reader to understand, the authors should provide detailed labelling of the figure notes. For example, Figure 2.

2. the authors used four typical anticancer drugs (Paclitaxel, Vincristine, Oxaliplatin and Bortezomib) for testing, please compare the advantages and disadvantages of the four drugs as well as their effects during the testing process.

3. in the conclusion section, the authors suggest more discussion of the novelty of the summarised work and insights into future work in related areas.

4. some important papers should be cited, such as Adv. Sci. 2023, 10, 2300601; Chemistry 2020, 2(3), 714-726; Angew. Chem. 2023, 135, e202306881.

Comments on the Quality of English Language

none

Author Response

  1. To make it easier for the reader to understand, the authors should provide detailed labelling of the figure notes. For example, Figure 2.

Thanks very much for your kind suggestion. Caption of Figure 2 in Page 5, Line 180 was modified as “Totally and individually changes of four electrical parameters (i.e., the impedance value, Coefficient of Variation (CV) of impedance value, total spikes, and CV of total spikes) in DRG after 24h, 72h, 168h exposure to A) control drug DMSO and B) negative drug sucrose. The upper column a) showed totally fold changes of four parameters compared to that before drug administration; data was shown as mean + standard deviation (SD), n = 5 MEA plates for both DMSO and sucrose. The lower column b) showed the distribution maps of the impedance value changes and total spikes changes in each individual MEA plate at different time-point. One table represent a MEA plate with 16 electrodes, and one grid in the table represent one electrode in the MEA plate; electrodes with an increasing value over 2 times of SD compared to the value before drug ad-ministration were marked as red, while electrodes with a decreasing value over 2 times of SD were marked as blue.” And captions of Figure 3~6 were also modified.

  1. the authors used four typical anticancer drugs (Paclitaxel, Vincristine, Oxaliplatin and Bortezomib) for testing, please compare the advantages and disadvantages of the four drugs as well as their effects during the testing process.

Thanks very much for your question. Among these four drugs, bortezomib showed a dose-dependent response in impedance changes. This could be an advantage of the current assessment method, since the mechanisms of bortezomib-induced neuropathy is not clear yet and there is no standard for detecting it.

  1. in the conclusion section, the authors suggest more discussion of the novelty of the summarized work and insights into future work in related areas.

Thanks very much for your kind suggestion. A sentence was added in Page 11, Line 325, as “In summary, the study contributes to advancing our understanding of CIPN mechanisms and highlights the utility of in vitro electrical assessments for predicting and assessing peripheral neurotoxicity. This research could have implications for the development of safer and more effective anti-cancer therapies by allowing for the early identification of potential neurotoxic effects during the drug development process.”

  1. some important papers should be cited, such as Adv. Sci. 2023, 10, 2300601; Chemistry 2020, 2(3), 714-726; Angew. Chem. 2023, 135, e202306881.

Thanks very much for your suggestion. Listed references were cited in the manuscript at Introduction in Page 1, and added into the Reference.